# Prognostic Value of C-Reactive Protein to Lymphocyte Ratio (CLR) in Emergency Department Patients with SARS-CoV-2 Infection

**DOI:** 10.3390/jpm11121274

**Published:** 2021-12-02

**Authors:** Ndenga Tonduangu, Pierrick Le Borgne, François Lefebvre, Karine Alame, Lise Bérard, Yannick Gottwalles, Lauriane Cipolat, Stéphane Gennai, Pascal Bilbault, Charles-Eric Lavoignet, Laure Abensur Vuillaume

**Affiliations:** 1Emergency Department, Regional Hospital of Metz-Thionville and France, 57000 Metz, France; liliapple@hotmail.fr (N.T.); cipolat@chr-metz-thionville.fr (L.C.); 2Faculty of Medicine, Lorraine University, 54500 Vandoeuvre-les-Nancy, France; 3Emergency Department, University Hospital of Strasbourg, 67000 Strasbourg, France; Pierrick.LEBORGNE@chru-strasbourg.fr (P.L.B.); karine.alame@chru-strasbourg.fr (K.A.); Pascal.Bilbault@chru-strasbourg.fr (P.B.); 4Regenerative NanoMedicine (RNM), Fédération de Médecine Translationnelle (FMTS), INSERM (French National Institute of Health and Medical Research), UMR 1260, Strasbourg University, 67000 Strasbourg, France; 5Department of Public Health, University Hospital of Strasbourg, 67000 Strasbourg, France; francois.lefebvre@chru-strasbourg.fr; 6Emergency Department, Haguenau Hospital, 67500 Haguenau, France; lise.berard@ch-haguenau.fr; 7Emergency Department, Colmar Hospital, 68026 Colmar, France; yannick.gottwalles@ch-colmar.fr; 8Emergency Department, Reims University Hospital, 51100 Reims, France; sgennai@chu-reims.fr; 9Emergency Department, Hôpital Nord Franche Comté, 90000 Belfort, France; charles-eric.lavoignet@hnfc.fr

**Keywords:** COVID-19, CLR, severity, mortality

## Abstract

(1) Introduction: According to recent studies, the ratio of C-reactive-protein to lymphocyte is more sensitive and specific than other biomarkers associated to systemic inflammatory processes. This study aimed to determine the prognostic value of CLR on COVID-19 severity and mortality at emergency department (ED) admission. (2) Methods: Between 1 March and 30 April 2020, we carried out a multicenter and retrospective study in six major hospitals of northeast France. The cohort was composed of patients hospitalized for a confirmed diagnosis of moderate to severe COVID-19. (3) Results: A total of 1,035 patients were included in this study. Factors associated with infection severity were the CLR (OR: 1.001, CI 95%: (1.000–1.002), *p* = 0.012), and the lymphocyte level (OR: 1.951, CI 95%: (1.024–3.717), *p* = 0.042). In multivariate analysis, the only biochemical factor significantly associated with mortality was lymphocyte rate (OR: 2.308, CI 95%: (1.286–4.141), *p* = 0.005). The best threshold of CLR to predict the severity of infection was 78.3 (sensitivity 79%; specificity 47%), and to predict mortality, was 159.5 (sensitivity 48%; specificity 70%). (4) Conclusion: The CLR at admission to the ED could be a helpful prognostic biomarker in the early screening and prediction of the severity and mortality associated with SARS-CoV-2 infection.

## 1. Introduction

Human coronavirus disease 2019 (COVID-19) is caused by a novel respiratory virus named severe acute respiratory syndrome coronavirus 2 (SARS-CoV-2), which first emerged in Wuhan, China, in December 2019, before resulting a global pandemic [1]. COVID-19 is now identified as a multi-systemic infection involving the hematological and immunological systems, responsible for a generalized inflammatory response due to the unregulated release of pro-inflammatory cytokines. This cytokine storm is responsible for the poor prognosis of COVID-19 patients [2]. Hence, biochemical parameters reflecting the immune and inflammatory status (such as monocytes, neutrophils, lymphocytes, interleukin-6, C-reactive protein (CRP), procalcitonin, platelets, D-dimer, erythrocyte sedimentation rate, ferritin, total bilirubin, and lactate dehydrogenase (LDH)) were under investigation for the disease severity assessment of COVID-19 [3,4,5]. Chang et al., Zhou et al., and Zhang et al. demonstrated in a multi-marker approach that in independent association, high neutrophilia (>7 × 10^3^/mm^3^), lymphopenia (<0.8 × 10^3^/mm^3^), significantly decreased CD4+ and CD8+ counts, elevated CRP (>47.5 mg/L), and elevated LDH (>593 U/L) were significant predictors of mortality, with odd ratios of 6.4 for neutrophils, 5.8 for CRP, and 4.2 for LDH [6,7,8,9,10,11,12,13].

In severe COVID-19, studies showed a pronounced elevation of CRP, indicating an increased systemic inflammatory response, and a decrease in lymphocytes, indicating a disruption of the immune response by the virus. Although CRP elevation and lymphopenia were not specific for COVID-19 severity and mortality assessment, CLR was found more useful than CRP or lymphocytes when considered separately [14]. Other biomarkers, such as ferritin, a protein reflecting macrophage inflammation in its acute stages, were judged more specific in current viral infections than CRP or lymphocytes alone [15,16]. Other biomarkers are considered specific for infection, such as interleukin 1 beta (IL-1) or Krebs von den Lungen-6 glycoprotein (KL-6) [17,18]. The latter two are certainly more specific to inflammation, but can be complex to use in a daily practice [15]. Moreover, CLR was previously identified as a risk factor for severity in cancer and infectious diseases, such as acute or perforated appendicitis [19,20]. Although several authors have found an association between CRP, lymphopenia, severity, and mortality in COVID-19, few have evaluated the CLR in this context [21,22,23]. The main objective of our study was to determine the prognostic value of CLR on COVID-19 severity and mortality at ED admission. 

## 2. Materials and Methods

### 2.1. Study Population and Settings

We conducted this retrospective multicentric study in six ED of the northeast of France. We led our study in two university hospitals (CHRU of Strasbourg, and CHU of Reims) and four general hospitals (Colmar Hospital, Nord Franche-Comté Hospital, Metz-Thionville Hospital, and Haguenau Hospital). These hospital centers, along with the entire greater-east region of France, were one of the outbreak’s epicenters in Europe during the first wave of the pandemic. We included all adult patients who were hospitalized for COVID-19 after presenting to the ED between 1 March and 30 April 2020. All patients in our study had a laboratory-confirmed diagnosis of COVID-19 by RT-PCR on nasopharyngeal swab. We excluded patients who had a non-confirmed diagnosis, those who received outpatient care, and those who received palliative therapy or limitation of therapeutic effort at their admission to the ED. Patients with a medical history or treatment that altered their blood counts and, therefore, their circulating lymphocytes or CRP (e.g., chemotherapy, immunosuppressive therapy, long- and short-term corticosteroid therapy, pre-admission antibiotic therapy, active cancer, or hematological malignancies) were also excluded from our study.

### 2.2. Data Collection

We retrospectively collected epidemiological, clinical, and biochemical data from patients’ electronic medical records, and standardized the results in a report file. We recorded symptom onset date along with the patient’s current treatment and medical history (including cardiovascular disease, diabetes, pre-existing renal failure, cancer, and hematological diseases). The primary endpoint was determining the prognostic value of CLR on COVID-19 mortality at ED admission. The secondary endpoint was determining its prognostic value on COVID-19 severity at ED admission. In this study, severe disease was defined by patient admission in to the ICU, which, during the first wave of the pandemic, was mainly associated with invasive mechanical ventilation indication. Moderate disease was defined by patient admission to conventional hospitalization units and, in fine, the requirement for simple or high flow oxygen therapy. Ambulatory patients were excluded. Obesity was defined by a body mass index superior to 30 kg/m^2^. Standard biochemical parameters were collected, such as levels of creatinine, CRP, and total leukocytes and lymphocytes. Lastly, we measured CLR values at ED admission, and the ratio of CRP to circulating lymphocytes. All collected data are summarized in the Tables and Results sections.

### 2.3. Ethics

This study was approved by the local ethics committee of the University of Strasbourg in France (reference CE: 2020–39), which, in accordance with the French legislation, waived the need for informed consent of patients whose data were entirely retrospectively studied.

### 2.4. Statistical Analysis

The descriptive analysis for categorical variables was executed by providing the frequency of each value. As for continuous variables, the analysis was done by giving median, first, and third quartiles. Wilcoxon tests were performed to compare the continuous covariates. Chi-squared tests or Fisher tests were performed to compare the categorical covariates in case the expected values in any of the cells of a contingency table was below 5. Using statistically significant results from univariate analyses and clinically relevant covariates, a multivariate logistic model was performed to assess in-hospital mortality, then disease severity. A backward stepwise method was performed based on AIC. ROC curves were performed to determine a threshold to discriminate between severe and moderate patients, and between patients who died during their stay and those who survived. Analyses were performed with the R software in version 4.0.2 (R Core Team 2020. R: A language and environment for statistical computing. R Foundation for Statistical Computing, Vienna, Austria), as well as with all the software packages required to carry out the analyses.

## 3. Results

### 3.1. Characteristics of the Study Population

During the study period, a total of 49,326 patients were admitted to the ED of all six centers combined. Among these patients, 4470 were diagnosed with SARS-CoV-2 infection, confirmed by PCR on nasopharyngeal swab. In total, 1035 patients were included in our study (Figure 1). 

Our cohort had a median age of 69 years (58–79 years) and was predominantly male (58.8%, CI 95%: 55.8–61.8%). Regarding comorbidities, one third of our study population was obese (34%), over half of the patients (56.7%) had high blood pressure, over a quarter of them (26.7%) had a history of diabetes, 23.2% of them had pre-existing renal failure, and only 1.7% had pre-existing liver failure. Over three-quarters (77.2%) of patients did not show any loss of functional autonomy. Regarding laboratory findings at admission to the ED, the median CLR was significantly higher in the group with severe disease compared to those with moderate disease (83.0, CI 95%: 33.3–173.5 versus 163.9, CI 95%: 83.8–310), *p* < 0.0001). Clinical and biochemical characteristics patients are summarized in Table 1. 

### 3.2. Biochemical Factors Associated COVID-19 Severity

Of the entire study population, 789 patients (76.2%) had moderate disease, whereas 246 (23.8%) had severe disease requiring ICU management. In multivariate analysis, adjusted on age, gender, complications, and laboratory findings after a backward stepwise selection, parameters associated with the severity of infection were lymphocytes (OR: 1.951, CI 95%: 1.024–3.717), *p* = 0.042), CLR (OR: 1.001, CI 95%: 1.000–1.002, *p* = 0.012), and CRP (OR: 1.009, CI 95%: (1.007–1.011), *p* < 0.001). These values are summarized in Table 2.

### 3.3. Factors Predicting COVID-19 Severity

We determined two ROC curves to predict the risk of disease severity. The area under the curve (AUC) for CLR at ED admission was 0.679 (CI 95%: 0.636–0.722), *p* < 0.001). The best cutoff for predicting the severity of infection was 78.3, with a sensitivity of 79% (95% CI: 72–85%), and a specificity of 47% (CI 95%: 44–50). In multivariate analysis, when adjusted on mortality, the OR of severity for a CLR greater than 78.3 was evaluated at 3.265 (CI 95%: 2.167–4.920, *p* = 0.0001). On admission to the ED, a high CLR (>78.3) was associated with significantly higher probabilities of admission to ICU (in univariate and multivariate analysis, respectively, OR: 3.4, CI 95%: 2.325–5.177, *p* < 0.0001; and OR: 3.265, CI 95%: 2.167–4.920, *p* < 0.0001). These results are presented in Figure 2. 

### 3.4. Biochemical Factors Associated with COVID-19 Mortality

A total of 139 patients died during their hospital stay, representing 13.4% (CI 95%: 11.4–15.5) of the cohort. In univariate analysis, the biomarkers CLR, CRP, and lymphocytes were associated with mortality at admission to the ED. However, lymphocyte levels were the only biochemical parameter significantly associated with mortality in the multivariate analysis (OR: 2.308, CI 95%: 1.286–4.141, *p* = 0.0051), and when adjusted on age, gender, complications, and laboratory findings after a backward stepwise selection, the CLR was not significant (OR: 1.001, CI95% 1.000–1.003, *p* = 0.090). These results are summarized in Table 3.

The AUC for CLR at ED admission was 0.607 (CI 95%: 0.554–0.659). The best CLR cutoff to predict the risk of death was 159.5, with a sensitivity of 48% (CI 95%: 3956), and a specificity of 70% (CI 95%: 67–73). The multivariate analysis adjusted on severity was considered significant, with an OR of mortality for a CLR greater than 159.5 evaluated at 1.952 (CI 95%: 1.333–2.857, *p* = 0.0006). These results are presented in Figure 3, and consistent with those of the validation cohort.

COVID-19 mortality was higher in patients with a CLR higher than 159.5 (respectively, in univariate and multivariate analysis, OR: 2.248, CI 95%: 1.553–3.255, *p* < 0.0001; and OR: 1.952, CI 95%: 1.333–2.857, *p* < 0.0006). 

## 4. Discussion

We studied the prognostic value of CLR in a cohort of patients with moderate and severe forms of COVID-19 at admission to the ED. We have shown that CLR is a relevant marker, significantly associated with the severity and mortality of COVID-19. However, used alone, it does not seem powerful enough to discriminate severity nor mortality. One of the major keys in the management of this pandemic is the control of the number of ICU patients, which has been the main reason for health care system oversaturation and disease mortality increase [24]. Therefore, it is essential to determine criteria allowing the referral of a COVID-19 patient in ICU.

Our results are consistent with other recent studies on this subject. Turan et al., in a cohort of 84 patients, studied the relationship between several biomarker ratios and the prognosis on the severity of COVID-19, its mortality, and the need for ICU [25]. The CLR was the only significant predictor of the three parameters studied (CLR, lymphocyte level, CRP level) in predicting disease severity (AUC = 0.766, *p* < 0.001, sensitivity 89.29%, specificity 53.57%), mortality (AUC = 0.696, *p* = 0.029, sensitivity 45.45%, specificity 90.41%), and ICU indication (AUC = 0.746, *p* < 0.001, sensitivity 92.31%, specificity 49.30%). Miao Yang et al. found similar results in a cohort of 108 patients [14]. Nevertheless, and unlike the other authors, we refined our cohort by excluding patients who might have medical history or treatment altering their CBC. Therefore, our results provide a further level of analysis on the effect of SARS-CoV-2 on white blood cell count.

This ratio has a better sensitivity during the acute phase of inflammation, as CRP levels increase earlier than neutrophilia or lymphopenia in acute inflammation regardless of its origin (infection, cancer, autoimmune disease) [26]. Thus, a high level of CLR can be considered an independent biomarker representing the initial stages of inflammation. In parallel, it is also important to note that although a ratio such as NLR is known to correlate with COVID-19 severity, this ratio can be falsely impacted by high-dose or falsely low corticosteroid therapy in an immunocompromised patient [27]. The CLR ratio, in these situations, is more reliable in predicting disease severity because it is not affected by the confounding factors mentioned above [26]. However, there are specific biomarkers, such as ferritin, a protein reflecting macrophage inflammation in its acute stages, which Chen et al. found to be abnormally elevated and associated with an increased risk of severe complications in COVID-19 patients [28]. On the other hand, IL-1, an inflammatory and autoimmune protein released by macrophages in the event of cell infection or necrosis, was up to twice as high in COVID-19 patients hospitalized in intensive care compared to healthy subjects. However, only the NYU Langone Medical Center in New York recommended dosing IL-1 when admitting patients to the emergency department, as it would be very unreliable due to an extremely short half-life, and the test has bad value prognosis if not dosed in a timely manner [15,17,29]. The KL6 has also shown potential as a prognostic biomarker of COVID-19 pneumonia because serum KL-6 concentrations were significantly higher in severe compared to non-severe patients [18]. These more specific biomarkers for inflammation would be relevant in the context of COVID-19, however, outside of research conditions, particularly for IL1 and KL6, the laboratories do not offer their assays in daily routine and they are, unlike the CLR, difficultly accessible and more expensive [15]. To better interpret the significance of these results, and facilitate their clinical applications, studies have suggested various thresholds for CLR, and a median value of 100 has often been described in severe SARS-CoV-2 infections [26,30]. In our study, the cut-off value for CLR was 78.3 regarding disease severity prediction, and 159.5 regarding mortality prediction. Ullah et al. described a CLR threshold of 100, over which value admission was correlated with the need for mechanical ventilation and increased risk of complications (OR: 2.5, CI 95%: 1.3–5.0, *p* = 0.01; OR: 2.9, CI 95%: 1.47–6.1, *p* = 0.004) [26]. In another study of 609 patients, Gemcioglu et al. studied the predictive value of 11 ratios for the severity of COVID-19, and the optimal cut-off value obtained from the CLR was 122.2, with a sensitivity and specificity for disease severity prediction of, respectively: 84% and 58% (CI 95%: 0.735–0.803; *p* < 0.0001) [30]. Similarly, Zeng et al. found a predictive threshold of 120 (OR: 7.14, CI 95%: 4.10–12.44, *p* < 0.001) for the severity of COVID-19, with moderate or severe patients and survivors presenting a gradual decrease in CLR after receiving medical treatment, whereas critical patients and non-survivors presented a ratio maintaining at high levels [13]. 

In our study, the CLR certainly has a very good sensitivity for predicting the severity of COVID-19, but its specificity for disease severity, and its sensitivity for mortality, lack power. To improve the prognostic performance of markers such as CLR in COVID-19, recent studies have proposed that they be incorporated with other epidemiological, clinical, and biological variables in a nomogram [31]. This process is taken from a technique called machine learning (ML), which uses the learning ability of an artificial intelligence to create an algorithm that is applied in the form of a prognostic score based on medical data from a patient [32,33]. In a study of 1955 patients, López-Escobar et al. developed four models of COVID-19 hospital mortality risk score (RIM score COVID-19), one of which is based on age, gender, oxygen saturation, CRP, and NLR, obtained an AUC of 0.853 (95% CI: 0.813–0.892), and was found useful in predicting the risk of death from COVID-19 at hospital admission [31]. Albarran-Sanchez et al. also offered a score combination of CLR and NLR (80% sensitivity and 74% specificity) to predict the morality of COVID-19 [34]. In view of these results, a global risk score model for COVID-19 integrating the CLR would seem to be an interesting avenue.

### Limitations

Our study has several limitations. Firstly, it is retrospective in nature, hence the data are subject to other confounding factors regardless of the number of exclusion criteria added (notably, comorbidities modifying the CBC). Secondly, our study data were collected during the first wave of the COVID-19 pandemic, when most patients were not receiving therapies altering lymphocyte levels, such as corticosteroids (one of the keystones in the treatment of severe COVID-19 today) or non-recommended antibiotics modifying the CBC.

## 5. Conclusions

The CLR appears to be a biomarker associated with COVID-19 severity and mortality. An isolated use of this biomarker seems unspecific. In future work, it could nevertheless be coupled with others in a multi-marker approach, or through machine learning technology for a combined severity prediction score. In effect, the diagnosis of COVID-19 could be automated and more accurate in early stages where, with the help of algorithms created by artificial intelligence, CLR could be integrated in a combination risk score.

## Figures and Tables

**Figure 1 jpm-11-01274-f001:**
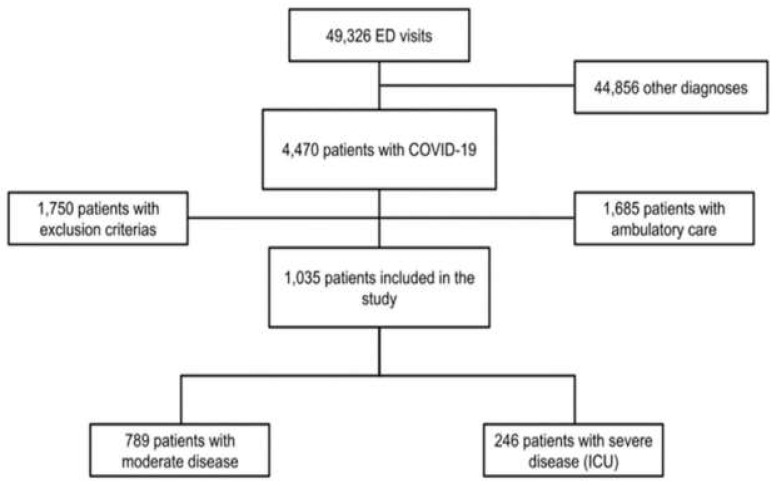
Flowchart of the study. Abbreviations: ED = emergency department; ICU = intensive care unit; COVID-19 = coronavirus disease 2019.

**Figure 2 jpm-11-01274-f002:**
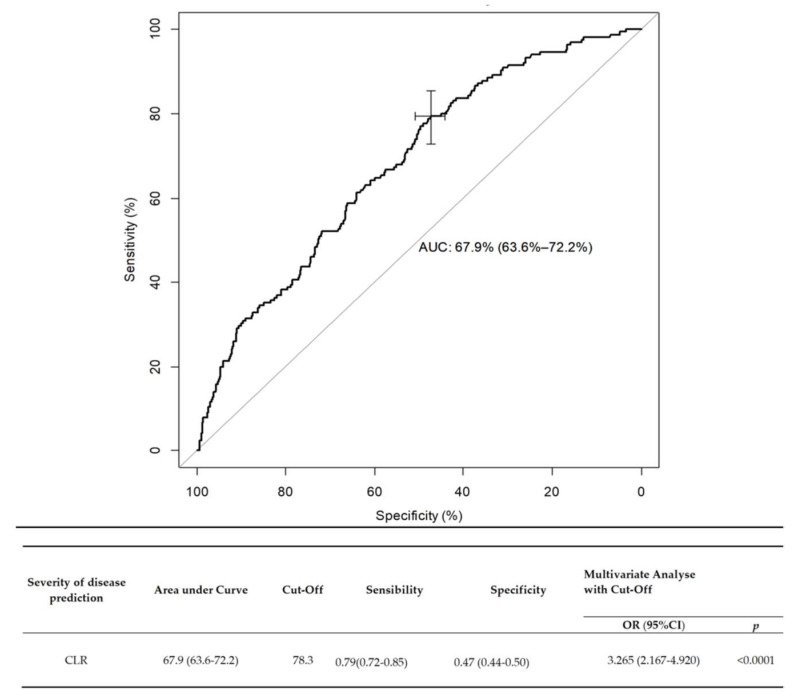
Receiver operating characteristics (ROC) curve of CLR as a predictive factor of COVID-19 severity. Data are expressed in median (Q1–Q3), Abbreviations: OR = odds ratio; CLR = C-reactive protein/lymphocyte count ratio.

**Figure 3 jpm-11-01274-f003:**
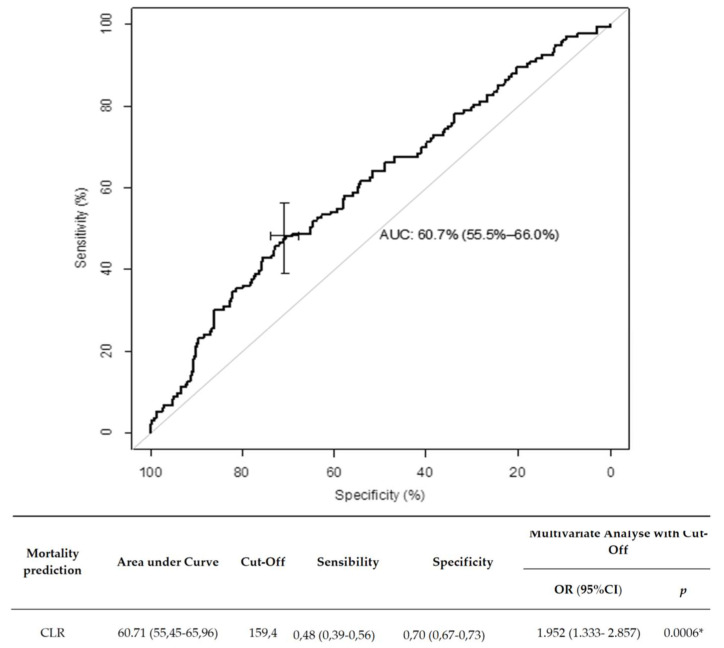
Receiver operating characteristics (ROC) curve of CLR as a predictive factor of COVID-19 mortality. Data are expressed in median (Q1–Q3), *: *p* < 0.005. Abbreviations: OR = odds ratio, CLR = C-reactive protein/lymphocyte count ratio.

**Table 1 jpm-11-01274-t001:** Demographics, baseline, and laboratory characteristics of moderate to severe COVID-19 patients.

	All Patients n = 1035	Moderate COVID-19 n = 789	Severe COVID-19 n = 246	*p*
Characteristics				
Age (years)	69 (58–79)	70 (58–81)	66 (57.3–72)	<0.001 *
Gender male	609 (58.8)	433 (54.9)	176 (71.5)	<0.001 *
Obesity	281 (36.9)	193 (35.0)	88 (41.9)	0.076
Comorbidities				
Hypertension	587 (56.7)	453 (57.4)	134 (54.5)	0.416
Diabetes mellitus	275 (26.7)	202 (25.6)	73 (26.6)	0.207
CKD	237 (23.2)	199 (25.5)	38 (15.8)	0.002 *
Cardiovascular disease	357 (34.5)	291 (36.9)	66 (26.8)	0.004 *
Total autonomy	796 (77.2)	569 (72.4)	227 (92.7)	<0.001 *
Respiratory disease	203 (19.6)	151 (19.1)	52 (21.1)	0.490
Laboratory Findings				
CRP (mg/L)	81 (39–142.3)	68 (33–128)	124 (76–192)	<0.001 *
Lymphocyte (×10^9^/L)	870 (630–1200)	900 (640–1220)	780 (590–1122)	0.003 *
CLR	97.0 (39.3–189.5)	83.0 (33.3–173.5)	163.9 (83.8–310)	<0.0001*
Outcome				
Hospital stay (days)	10 (7–17.3)	8 (6–12)	24 (17–38)	<0.001 *
Intra-hospital mortality	139 (13.6)	82 (10.4)	57 (24.1)	<0.001 *

Data are all expressed in median (Q1–Q3) or n/N (%), where N is the total number of patients with available data. BMI = body mass index, CKD = chronic kidney disease, CRP = C-reactive protein, CLR = CRP/lymphocyte count ratio * *p* < 0.05.

**Table 2 jpm-11-01274-t002:** Univariate and multivariate analysis of risk factors for COVID-19 severity.

	All	Moderate	Severe	% Missing Data	Univariate		Multivariate	
					Analysis		Analysis	
					OR (95% CI)	*p*	OR (95% CI)	*p*
Lymphocytes (×10^9^/L)	870 (620–1200)	890 (630–1210)	870 (620–1200)	1.5	0.864 (0.618–1.209)	0.3950	1.951 (1.024–3.717)	0.0422 *
CRP (mg/L)	81 (39–142)	71 (35–131)	129.0 (76.0–195.0)	0.7	1.008 (1.006–1.010)	<0.0001 *	1.009 (1.007–1.011)	<0.0001 *
CLR	97 (39.3–189.5)	83 (33.3–173.5)	163.9 (83.8–310.0)		1.002 (1.001–1.003)	<0.0001 *	1.001 (1.000–1.002)	0.0120 *

*: *p* < 0.005. Abbreviations: OR = odds ratio; CRP = C-reactive protein; CLR = C-reactive protein/lymphocyte count ratio.

**Table 3 jpm-11-01274-t003:** Univariate and multivariate analysis of risk factors for COVID-19 mortality.

	Alive n = 884	Died n = 139	Univariate Analysis		Multivariate Analysis	
			OR (95% CI)	*p*	OR (95% CI)	*p*
Lymphocytes (×10^9^/L)	0.89 (0.65–1.22)	0.72 (0.50–1.00)	0.524 (0.336–0.815)	0.0042 *	2.308 (1.286–4.141)	0.0051 *
CRP (mg/L)	78.5 (37.0–139.0)	100.0 (56.0–158.0)	1.003 (1.001–1.005)	0.0065 *	1.000 (0.997–1.004)	0.814
CLR	90.5 (36.0–177.3)	136.4 (54.4–259.6)	1.002 (1.001–1.002)	0.0001 *	1.001 (1.000–1.003)	0.090

*: *p* < 0.005. Abbreviations: OR = odds ratio; CRP = C-reactive protein; CLR = C-reactive protein/lymphocyte count ratio.

## Data Availability

The data presented in this study are available on request from the corresponding author.

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
