# Peer review of "Prognostic Value of C-Reactive Protein to Lymphocyte Ratio (CLR) in Emergency Department Patients with SARS-CoV-2 Infection"

_jpm, 2021, doi:10.3390/jpm11121274_

Round 1

Reviewer 1 Report

The prognosis of the Covid-19 may range from complete well-being to severe acute respiratory distress syndrome or death. Therefore, finding the easily applicable index to determine the severity of the diseases at the presentation is well worth the effort. Even though the topic of the manuscript is very interesting and touches currently important issue I have few objection to the study.

  1. The authors found the CLR index as a relevant parameter which predict the severity of disease. However, the accuracy of a diagnostic test is the traditional academic point system, which indicates that CLR index would be considered to be "poor" at separating moderate and severe disease or fatality of the disease. The areas under the CLR ROC curves were 0.61-0.68.
  2. I cannot find any reason for Table 4 presentation. This odd analysis shows severity and mortality of the disease as independent predictors of different CLR cutoff values. It is uncommon and not right assumption of statistic analysis. In my opinion, this Table and analysis should be deleted.
  3. Figs of ROC analysis are too big.
  4. It is generally accepted to use the shortcut "et al." instead of "and al".
  5. In the title of the manuscript abbreviation "ED" should be expanded.

Author Response

#Reviewer 1

Open Review

English language and style

( ) Extensive editing of English language and style required
( ) Moderate English changes required
(x) English language and style are fine/minor spell check required
( ) I don't feel qualified to judge about the English language and style

Yes

Can be improved

Must be improved

Not applicable

Does the introduction provide sufficient background and include all relevant references?

(x)

( )

( )

( )

Is the research design appropriate?

(x)

( )

( )

( )

Are the methods adequately described?

(x)

( )

( )

( )

Are the results clearly presented?

( )

( )

(x)

( )

Are the conclusions supported by the results?

(x)

( )

( )

( )

Comments and Suggestions for Authors

The prognosis of the Covid-19 may range from complete well-being to severe acute respiratory distress syndrome or death. Therefore, finding the easily applicable index to determine the severity of the diseases at the presentation is well worth the effort. Even though the topic of the manuscript is very interesting and touches currently important issue I have few objection to the study.

  1. The authors found the CLR index as a relevant parameter which predict the severity of disease. However, the accuracy of a diagnostic test is the traditional academic point system, which indicates that CLR index would be considered to be "poor" at separating moderate and severe disease or fatality of the disease. The areas under the CLR ROC curves were 0.61-0.68.

Response: We agree with this observation. We modified our manuscript nuancing our claims with a degree of moderation. As explained at the end of the discussion, this marker is not powerful enough on its own and needs to be integrated in a more global approach. We hope this new version will meet your expectations.

  1. I cannot find any reason for Table 4 presentation. This odd analysis shows severity and mortality of the disease as independent predictors of different CLR cutoff values. It is uncommon and not right assumption of statistic analysis. In my opinion, this Table and analysis should be deleted.

Response: We used the cut-off values found in literature as well as the cut-off value from our study. However, we agree that it is an uncommon approach and have followed by removing Table 4.

  1. Figs of ROC analysis are too big.

Response: We have reduced the size of the figures.

  1. It is generally accepted to use the shortcut "et al." instead of "and al".

Response: We thank the reviewer for this remark, the translation was corrected accordingly.

  1. In the title of the manuscript abbreviation "ED" should be expanded.

Response: We expanded on the abbreviation accordingly.

Reviewer 2 Report

Comments to the Author

The author suggested that The CLR could be a helpful prognostic biomarker in the early screening and prediction of severity and mortality in COVID-19 patients. The present study has clinical significance and relatively large number of subjects.  However, sufficient improvements for those issues are needed for publication on Journal of Personalized Medicine.

Major comments

  1. In univariate and multivariate analysis using logistic model, only CRP, lymphocyte count, and CLR are analyzed. Did you perform multivariate analysis including age, gender, and complications, which were factors significantly associated in the present study? Otherwise, did you analyze the effects of those factors on the CLR? Please perform at least one of the above analyses and consider the influence of the results on the main opinions in this study.

  1. In the present study, the subjects who have moderate or higher severity, so CLR may not be a useful predictor of severity in early screening. In order to say that, I consider that analyses of the patient group including mild cases. Please reconsider the description of biomarkers useful for early screening.

  1. Increase of CRP level and decrease of lymphocyte count are considered to be nonspecific biomarkers of inflammation and viral infections, but specific biomarkers such as ferritin reflecting macrophage inflammation, IL-1b reflecting NETosis, and KL-6 reflecting pulmonary fibrosis in COVID-19 patients. Also discuss those biomarkers.

  1. Please show if any patients had liver dysfunction that could influence to CRP levels or lymphocyte counts.

Author Response

#Reviewer 2

Open Review

English language and style

( ) Extensive editing of English language and style required
(x) Moderate English changes required
( ) English language and style are fine/minor spell check required
( ) I don't feel qualified to judge about the English language and style

Yes

Can be improved

Must be improved

Not applicable

Does the introduction provide sufficient background and include all relevant references?

( )

(x)

( )

( )

Is the research design appropriate?

( )

( )

(x)

( )

Are the methods adequately described?

( )

( )

(x)

( )

Are the results clearly presented?

( )

(x)

( )

( )

Are the conclusions supported by the results?

( )

( )

(x)

( )

Comments and Suggestions for Authors

Comments to the Author

The author suggested that The CLR could be a helpful prognostic biomarker in the early screening and prediction of severity and mortality in COVID-19 patients. The present study has clinical significance and relatively large number of subjects.  However, sufficient improvements for those issues are needed for publication on Journal of Personalized Medicine.

 Major comments

  1. In univariate and multivariate analysis using logistic model, only CRP, lymphocyte count, and CLR are analyzed. Did you perform multivariate analysis including age, gender, and complications, which were factors significantly associated in the present study? Otherwise, did you analyze the effects of those factors on the CLR? Please perform at least one of the above analyses and consider the influence of the results on the main opinions in this study.

Response: The multivariate analysis was performed using statistically significant results from univariate analyses and clinically relevant covariates. A backward stepwise method based on AIC was used to select the final model in which age, gender, and complications were included. Only the parameters of lymphocytes, CRP and CLR were presented. The manuscript has been amended to clarify this point.

  1. In the present study, the subjects who have moderate or higher severity, so CLR may not be a useful predictor of severity in early screening. In order to say that, I consider that analyses of the patient group including mild cases. Please reconsider the description of biomarkers useful for early screening.

Response: We thank the reviewer for this remark. True, we did not include patients who were initially non moderate or severe, and those who would have deteriorated afterwards. Our analysis is on the initial triage in the emergency room during a pandemic. To date, the perfect biomarker does not exist. KL-6 and IL-1b have excellent predictive values, but are not fit for use in daily routine. The CLR has a real clinical application in the emergency department.t Although it cannot be used alone, it can help physicians.

  1. Increase of CRP level and decrease of lymphocyte count are considered to be nonspecific biomarkers of inflammation and viral infections, but specific biomarkers such as ferritin reflecting macrophage inflammation, IL-1b reflecting NETosis, and KL-6 reflecting pulmonary fibrosis in COVID-19 patients. Also discuss those biomarkers.

Response: Certainly, CRP, ferritin, lymphocytes, neutrophils would are relevant in the context of COVID-19, complementing the clinical picture in severe to critical stages. These markers allow better risk stratification for prognosis and guide clinical decisions by identifying patients requiring increased surveillance or adapted treatment. The glycoprotein KL-6 reflects the severe interstitial lung injury, pulmonary epithelial changes and fibrosis processes secondary to SARS-CoV infection and has shown potential as a prognostic biomarker for COVID-19 pneumonia since its serum level was only elevated in patients with severe lung injury and increased risk of ARDS or intubation. The IL-1 assay may have added value in a research protocol, but at present it has no clinical utility that would result in different management of patients with COVID-19. We have modified the manuscript to incorporate these data and thank the reviewer for their suggestions.

  1. Please show if any patients had liver dysfunction that could influence to CRP levels or lymphocyte counts.

Response : We, unfortunately, were not able to present the liver enzymes of the patients at admission, as ED patients did not have this biochemical analysis systematically performed during the first wave of the pandemic. However, we have the medical history of pre-existing liver failure, which only concerns 1.7% of patients, indicating that most patients had a healthy liver, which does not influence our CRP results, and we suggest adding this element in the revised manuscript. If the reviewer wishes, we can also add the following parameters as additional material: bilirubin, PT, platelets, yet neither the markers of cytolysis nor those of cholestasis were collected.

Round 2

Reviewer 2 Report

Author have correctly responded to my comment.